# Genomic Diversity of SARS-CoV-2 Omicron Sublineages and Co-Circulation with Respiratory Viruses in Pediatric Patients in Sao Paulo, Brazil

**DOI:** 10.3390/v17111421

**Published:** 2025-10-25

**Authors:** Erick Gustavo Dorlass, Guilherme Pereira Scagion, Fabyano Bruno Leal de Oliveira, Bruna Larotonda Telezynski, Ana Karolina Antunes Eisen, Giovana Santos Caleiro, Isabela Barbosa de Assis, Camila Araújo Valério, Vanessa Nascimento Chalup, Cairo Monteiro de Oliveira, Camila Ohomoto de Morais, Marcelo Otsuka, Vera Bain, Mariana Pereira Soledade, Luciano Matsumiya Thomazelli, Carolina Sucupira, Luciana Becker Mau, Andressa Simões Aguiar, Flávia Jacqueline Almeida, Marco Aurélio Palazzi Safadi, João Renato Rebello Pinho, Danielle Bruna Leal de Oliveira, Jansen de Araujo, Edison Luiz Durigon

**Affiliations:** 1Laboratório de Virologia Clínica e Molecular, Instituto de Ciências Biomédicas, Universidade de São Paulo, São Paulo 05508-000, Brazil; gui.scagion@gmail.com (G.P.S.); fabyo.leal@gmail.com (F.B.L.d.O.); brunatelelopes@gmail.com (B.L.T.); anaeisen@usp.br (A.K.A.E.); assisisabela.barbosa@gmail.com (I.B.d.A.); camilavalerio@alumni.usp.br (C.A.V.); vmnchalup@gmail.com (V.N.C.); cairomonteiro00@gmail.com (C.M.d.O.); lucmt@usp.br (L.M.T.); danielle.durigon@einstein.br (D.B.L.d.O.); 2Einstein Hospital Israelita, São Paulo 05652-900, Brazil; joao.pinho@einstein.br; 3Laboratório de Pesquisa em Vírus Emergentes, Instituto de Ciências Biomédicas, Universidade de São Paulo, São Paulo 05508-000, Brazil; giovanacaleiro@gmail.com; 4Departamento de Pediatria, Faculdade de Ciências Médicas da Santa Casa de São Paulo, São Paulo 01221-020, Brazil; camilaohomoto@gmail.com (C.O.d.M.); flaviaja@gmail.com (F.J.A.); masafadi@uol.com.br (M.A.P.S.); 5Serviço de Infectologia do Hospital Infantil Darcy Vargas, São Paulo 05614-040, Brazil; lageotsuka@yahoo.com.br (M.O.); ccih.infectoped@gmail.com (C.S.); 6Instituto da Criança da Faculdade de Medicina da Universidade de São Paulo, São Paulo 05403-000, Brazil; verinha.bain@gmail.com; 7Hospital Municipal Infantil Menino Jesus, São Paulo 01329-010, Brazil; lubeckermau@gmail.com; 8Hospital Infantil Candido Fontoura, São Paulo 03173-010, Brazil; ccih.nhe@gmail.com (M.P.S.); andressa.simoes@icb.usp.br (A.S.A.); 9Hospital São Luíz Gonzaga, São Paulo 02276-140, Brazil

**Keywords:** Omicron, SARS-CoV-2, respiratory virus, RSV, pediatric, infection

## Abstract

The SARS-CoV-2 Omicron variant caused a global surge in COVID-19 cases following its emergence in November 2021, rapidly diversifying in the subsequent months. Although many studies have documented Omicron’s diversification, few have explored its impact on pediatric populations or the seasonality of other respiratory viruses in children. This study aims to investigate the diversity and circulation patterns of SARS-CoV-2 Omicron sublineages in pediatric patients in São Paulo, Brazil, and assess their co-circulation with other respiratory pathogens. Respiratory samples collected from patients under 18 years old across five hospitals between January 2022 and April 2023 were tested for different respiratory viruses using real-time RT-PCR. Whole-genome sequencing was performed on SARS-CoV-2-positive samples. Among the 7868 pediatric respiratory samples tested, 3902 were positive for viral pathogens. Respiratory Syncytial Virus accounted for the highest number of positive cases (*n* = 1248), exhibiting an atypical off-season peak in November 2022. SARS-CoV-2 was detected in 297 samples, of which 103 were sequenced. BA.1 and BA.5 sublineages had predominant genomic diversity and circulation time. These findings highlight the Omicron variant’s significant impact on the epidemiology and seasonal distribution of respiratory viruses in children, emphasizing the ongoing need for vaccination and robust surveillance efforts in pediatric populations.

## 1. Introduction

Since its first detection in late 2019, SARS-CoV-2 has infected more than 700 million people and caused 7 million deaths worldwide, with over 700,000 deaths in Brazil [1,2,3]. Over time, the virus acquired genomic mutations that resulted in the emergence of novel variants with increased transmissibility and enhanced immune evasion, reducing the effectiveness of prior infection or vaccine-induced immunity [4].

The Omicron variant (B.1.1.529) was first identified in South Africa in November 2021 and was detected in Brazil within the same month [5,6], followed by a global increase in SARS-CoV-2 infection cases [7,8,9].

Omicron carries multiple mutations compared to previous Variants of Concern (VOCs), such as Alpha, Gamma, and Delta [6], and became dominant worldwide by the first semester of 2022 due to its increased infection rate and vaccine-breakthrough capabilities [10]. Different Omicron sublineages alternated dominance over time, such as BA.1, BA.2, BA.5, BQ.1.1, XBB.1.16, EG.5.1, JN.1, and KP.2 [10,11]. Despite the high prevalence of SARS-CoV-2 infections following the emergence of Omicron, other viruses continued to circulate [12], although with notable shifts in seasonal patterns [13,14].

While the evolutionary dynamics of the Omicron variant have been well documented [10,15,16], few studies have addressed Omicron’s genomic variation specifically in pediatric populations or investigated how the SARS-CoV-2 circulation has impacted other respiratory viruses in Brazil. In this study, we analyzed the genomic prevalence of the SARS-CoV-2 Omicron variant among pediatric patients during January 2022 and April 2023. Additionally, we assessed how Omicron’s emergence influenced the prevalence of other respiratory viruses in this population and period.

## 2. Materials and Methods

### 2.1. Sample Collection

Nasal-Oral swabs and nasopharyngeal aspirates were collected from patients under 18 years old with respiratory symptoms from 5 hospitals in the city of São Paulo, Brazil, Hospital Infantil Menino Jesus (HIMJ), Hospital São Luiz Gonzaga (HSLG), Hospital Cândido Fontoura (HCF), Hospital Infantil Darci Vargas (HICF), and Santa Casa da Misericórdia (STA), from January 2022 to April 2023.

### 2.2. Virus Detection

Total RNA was extracted using the MagMAX^TM^ Total Nucleic Acid Isolation Kit (Applied Biosystems, Waltham, MA, USA) using MagMAX Express—96. The extracted RNA was submitted to real-time RT-PCR using the QuantStudio 3 system (Thermo Fisher, Waltham, MA, USA) with the Allplex kit (Seegene Inc., Seoul, South Korea) for detection of different respiratory viruses: Respiratory Syncytial Virus (RSV), Metapneumovirus (MPV), Adenovirus (AdV), Human Influenza A and B (FluA and FluB), Human Parainfluenzavirus 1 to 4 (HPIV 1–4), Human Coronavirus (HCoV HKU, NL93, OC43 and 229E), Human Bocavirus (HBoV), Enterovirus, and SARS-CoV-2. A primers list is shown in Appendix A [17,18,19,20,21,22,23,24,25,26].

### 2.3. SARS-CoV-2 Whole Genome Sequencing

SARS-CoV-2-positive samples with Ct values lower than 30 were submitted to whole-genome sequencing. Extracted material was first quantified with Quibit4 (Invitrogen, Eugene, OR, USA) and was then submitted to cDNA synthesis with the SuperScript VILO Master Mix (Thermo Fisher Scientific, Waltham, MA, USA). Library preparation occurred in an amplicon-based automated method using Ion Chef (Thermo Fisher Scientific). Arctic V4 primers were used for amplification. Amplicons were fragmented to 225 bp. The assembly libraries were quantified with RT-qPCR, deposited in Ion Chip’s 530 using Ion Chef, and finally submitted to sequencing in Ion Torrent S5 (Thermo Fisher Scientific).

### 2.4. Genomic Analysis

Read files were generated with Torrent Suite v5.12 plugins on Ion Torrent S5 platform. Low-quality reads were removed from the analysis. Remaining reads were mapped directly to the SARS-CoV-2 reference genome (accession: MN908947) with TMAP. Coverage metrics were collected with the CoverageAnalysis plugin (v5.10) set for a minimum of 50 reads depth as previously described [27]. Variants were called with variantCaller plugin (v5.12) with default “Germ Line-Low Stringency” parameters, as seen in [28,29]. Called variants were annotated with COVID19AnnotateSnpEff (v1.0), and final consensus sequences were generated using the GenerateConsensus plugin. SARS-CoV-2 lineages were identified using Pangolin software (v4.3.1) [30].

### 2.5. Data Analysis

Descriptive statistics were used for frequency and distribution analyses of detected respiratory viruses across sampling months. Associations between categorical variables, such as virus occurrence and sampling period, were evaluated using the Chi-square test of independence, with *p*-values < 0.05 considered statistically significant. Data visualization and exploratory analysis were conducted using Python 3 by using the Pandas library (v2.1).

## 3. Results

A total of 7868 samples were tested for respiratory viruses from January 2022 to April 2023 by real-time RT-PCR. Of these, 3369 (42.81%) were positive for a single virus, and 533 (6.77%) were co-infections of two or more viruses, resulting in a total of 3902 (49.59%) positive samples for one or more viruses.

RSV accounted for the highest number of single-infection samples (*n* = 1248, 17.01%), followed by AdV (*n* = 435, 5.93%), HBoV (*n* = 324, 4.41%), HPiV (*n* = 330, 4.49%), SARS-CoV-2 (*n* = 297, 4.04%), MPV (*n* = 266, 3.62%), Enterovirus (*n* = 195, 2.65%), HCoV (*n* = 173, 2.35%), and FluA/B (*n* = 101, 1.37%) (Appendix A).

The periods from March to May 2022 and from September to November 2022 recorded the highest numbers of positive samples for a single virus, with averages of 255.3 and 255, respectively. Between June and September 2022, a reduction was observed in the number of positive samples for RSV, MPV, AdV, HPiV, and SARS-CoV-2. In contrast, the number of detections for HCoV and Enterovirus increased during the same period. The detection rates of HBoV and FluA/B remained relatively stable. The average number of positive samples recorded from June to September for a single virus was 239. February was the month with the lowest number of positive samples in 2022 (*n* = 116) and January in 2023 (*n* = 64) (Appendix A).

In 2022, RSV showed increased detection in March and October, reaching its highest number of positive samples in November (*n* = 186). AdV cases also rose in March, peaking in June with 92 detections. HBoV reached its highest detection rate in October (*n* = 43). MPV peaked in April (*n* = 61), and HPiV peaked in May (*n* = 63). HCoV showed a peak in August (*n* = 45), while Enterovirus was predominantly detected in March (*n* = 29). FluA/B exhibited low detection levels from February to August, followed by an increase in September and a peak in October (*n* = 15). SARS-CoV-2 reached its highest number of detections in January (*n* = 123), with additional increases observed in June and November 2022 (Figure 1). In 2023, most viruses peaked in April, excepting AdV, HBoV, Enterovirus, and SARS-CoV-2, which reached the highest number of cases in February (AdV and SARS-CoV-2) and March (HCoV and Enterovirus). RSV was the most prevalent virus in 2023, with a total of 294 detected samples that year, followed by HPiV (*n* = 75) and FluA/B (*n* = 61).

Co-infections involving RSV, AdV, and HBoV were the most frequent, with RSV-HBoV presenting the most cases of co-infection (*n* = 66), followed by RSV/AdV (*n* = 61) and HBoV/AdV (*n* = 59). These co-infections accounted for 36.77% (*n* = 196) of the total number of detected co-infections. RSV/SARS-CoV-2 were the fourth most common co-infection observed (*n* = 39, 7.31%). HCoV/Enterovirus, Flu/HPiV, Flu/RSV, and MPV/Enterovirus were the least frequent co-infections detected during the study. Co-infections involving two viruses accounted for 472 (88.55%) of the total co-infections. The most common co-infection caused by three viruses detected is AdV/HBoV/RSV (*n* = 12, 2.25%). Co-infection with four viruses were rare, accounting for five cases in total (0.75%, Appendix A). Cases peaked in June 2022 (*n* = 94) followed by October 2022 (*n* = 74). SARS-CoV-2 co-infections were most frequent in January 2022. However, June 2022 showed a higher number of types with SARS-CoV-2, such as RSV/SARS-CoV-2, HBoV/SARS-CoV-2, MPV/SARS-CoV-2, HPiV/SARS-CoV-2, AdV/SARS-CoV-2, HCoV/SARS-CoV-2, and HBoV/HPiV/SARS-CoV-2.

Single RSV infections exhibited a greater dispersion in detected samples compared to other viruses in the study, ranging from a minimal of 22 cases to 186. Overall, other respiratory viruses showed a low median value and narrower distribution in comparison to RSV, such as FluA/B (Figure 2).

Of the 297 total pediatric samples that tested positive exclusively for SARS-CoV-2, 103 (Ct < 30) were selected for whole-genome sequencing (Figure 3). January 2022 had the highest number of sequenced samples (*n* = 42), followed by June (*n* = 17) and November (*n* = 12). These months corresponded to those with the highest number of SARS-CoV-2 positive cases: January (*n* = 123), June (*n* = 25), and November (*n* = 28) (*p* < 0.05).

A total of 20 Omicron sublineages were identified. Among the 103 sequenced samples, BA.1 was the most prevalent, accounting for 22.5% of cases, followed by BA.4 (13.7%). The least frequent sublineages—FE.1.1, XBB.1.5, BQ.1.1.4, BQ.1.1.5, and XBB.1.18—were each detected in only 0.9% of samples. In 2023, XBB.1.5.86 emerged as the most prevalent sublineage (Figure 4).

BA.1 and BA.2 predominated between January and February 2022, until the emergence of BA.4 and BA.5, which subsequently became dominant. BA.5 remained prevalent until the detection of XBB.1.5 in 2023, regardless of the emergence of BQ.1 in the second semester of 2022. Among all, BA.5 was the most persistent, circulating from July to December 2022.

## 4. Discussion

A substantial proportion (49.59%) of pediatric samples tested positive for respiratory viruses, reflecting the high prevalence of these pathogens among children and adolescents presenting respiratory symptoms. RSV was the most frequently detected virus in this study, consistently appearing as a predominant pathogen throughout the surveillance period.

RSV is a major pathogen in young children, particularly relevant for infants and premature neonates, as it represents a leading cause of lower respiratory tract illness, hospitalization, and substantial morbidity in these vulnerable groups [31]. RSV seasonality is well established in the Southeast region of Brazil, characterized by an increase in cases starting in early autumn (March) and declining by winter (August) [32]. Our study observed the expected increase in the number of RSV cases beginning in March 2022, but also notably documented an atypical second increase starting in September of the same year, reaching a pronounced peak in November, accounting for the highest number of RSV-positive samples in 2022 (Figure 1, Appendix A).

Other studies have similarly documented a shift in RSV seasonality during the COVID-19 pandemic [13,14,33,34]. Our findings specifically highlight the impact of the SARS-CoV-2 Omicron variant’s emergence on RSV circulation, resulting in a distinct second wave of cases within the same year—an unusual pattern also observed in other southern hemisphere countries [35]. Interestingly, the observed second wave of RSV infections surged after a significant drop in the detection of SARS-CoV-2 in September and October (Figure 1). This was also observed in other studies with RSV [36] and other respiratory viruses [37,38], where an increase in respiratory infection cases took place after flexibilization or remotion of nonpharmaceutical intervention (NPI) measures. This highly correlates with our findings, since the second wave of RSV infections occurred after the end of mid-year recess of schools and our cohort constitutes only pediatric patients. In addition, the city of São Paulo implemented increased flexibility in NPI measures in September 2022. Moreover, many studies suggested that different RSV lineages were responsible for the increase in RSV cases in North America [39,40]. This emphasizes that, even during pandemic periods, the evolutionary dynamics of respiratory viruses, especially RNA viruses, continue to shape their epidemiological patterns, with impacts on pediatric populations.

Although other viruses such as HPiV and Enteroviruses showed a secondary increase in detected cases during the second semester of 2022, these variations were less abrupt compared to Respiratory Syncytial Virus. RSV demonstrated a sharp rise, with an 84.14% increase in positive cases from April to November 2022 in single cases, following an even steeper 114% increase between January and April 2022. This pronounced fluctuation, combined with the consistently higher number of cases, underscores the distinct epidemiological dynamics of RSV throughout 2022 (Figure 2).

Co-infections overall exhibited a similar trend to single infections, with an increase in the number of samples with two or more viruses observed in the first and second semesters of 2022 (March and October). A peak in the number of co-infection cases was observed in June 2022 (*n* = 92), which coincided with a surge in AdV detections (Figure 1, Appendix A). HBoV, AdV, and RSV accounted for most co-infection cases (Appendix A), which collaborates with other studies [41,42,43].

RSV was the virus most detected in co-infections (*n* = 272), particularly in October and November 2022, as a rise in single-infection RSV cases also was observed (Figure 1). SARS-CoV-2 showed a total of 79 co-infection cases during the study, with a majority detected in January 2022, (*n* = 16), June 2022 (*n* = 14), and November 2022 (*n* = 12). These months also correlate with a higher detection of single SARS-CoV-2-positive samples (Figure 3).

The total number of samples sequenced directly mirrored the monthly fluctuations in total SARS-CoV-2 detections, correlating closely with epidemiological waves driven by the introduction and spread of new sublineages in June 2022, November 2022, and February 2023 (Figure 3 and Figure 4).

The highest number of SARS-CoV-2-positive samples occurred in January 2022, coinciding with the emergence of the Omicron variant, first identified in Brazil in late November 2021. This period also coincided with a reduction in the circulation of other respiratory viruses, an observation consistent with findings from other studies associating high Omicron prevalence with reduced detection of other viral pathogens [12,44]. Interestingly, FluA/B showed a higher number of positive samples in January 2022 compared to its overall circulation that year. This can likely be attributed to the emergence of the Influenza A H3N2/Darwin strain in the end of 2021 [38].

During the first half of 2022, the Omicron sublineage BA.1 and several related BA.1 sublineages (including BA.1.1, BA.1.14, BA.1.14.2, and BA.1.15) were detected. The observed genetic diversity is likely attributable to the high circulation and prevalence of BA.1 during this period, which led to the continuous emergence of multiple subvariants [45]. This diversification persisted until the subsequent introduction of the BA.4 and BA.5 sublineages.

The BA.4 and BA.5 sublineages were first identified in Brazil in May 2022. These sublineages have been associated with increased immune evasion capabilities [46], possibly explaining the 103% increase in SARS-CoV-2 detections observed between May and June 2022.

The BA.5 sublineage demonstrated a prolonged circulation. It was identified continuously from June to December 2023. This long circulation period can justify the reason for BA.5’s genomic diversity, with a similar number of BA.5-related sublineages detected at the time of its circulation as BA.1 (Figure 4), including BE.9, BE.10, BA.5.2.1, and DL.1. This result correlates with other analysis of SARS-CoV-2 genomes sequenced in Brazil in 2022 [16].

The VOCs BQ.1 and XBB.1.5 were detected in November 2022 and February 2023, respectively, and different related sublineages were identified, including BQ1.1, BQ.1.1.4, BQ.1.1.5, BQ.1.1.22, XBB.1.15, XBB.1.18, and FE.1.1. These lineages showed significant immune evasion from antibodies, induced either by vaccination or prior infection, as shown in many studies [47,48,49]. Interestingly, our results show the dominance of the BA.5 sublineage from October to December 2022, only shifting after the detection of the recombinant XBB sublineage. This finding diverges with the analyses performed by Souza et al. (2025) [16], where counting the sequenced samples of Brazil in GISAID showed a high prevalence of BQ.1.1 sequences during 2022; however, this can be attributed to the limited number of detected and sequenced samples collected during the second semester of 2022.

While Omicron infections in children have generally presented with milder clinical severity compared to previous circulating variants such as Delta [50], hospitalized cases still carry a significant risk of developing severe disease [51]. Vaccination has proven to be an effective measure to prevent severe cases [52,53]. Our findings strongly support continued pediatric vaccination efforts, underscoring that SARS-CoV-2 can rapidly diversify within pediatric populations, facilitating the emergence of new sublineages.

Our data demonstrates significant genomic diversity among SARS-CoV-2 Omicron sublineages that circulated in pediatric patients between January 2022 and April 2023, underscoring the rapid evolutionary dynamics of the virus within this population. The frequent emergence and turnover of distinct sublineages highlight the critical need for ongoing, robust genomic surveillance systems to promptly detect and monitor viral variants. Moreover, the observed influence of Omicron circulation on the seasonal patterns of other respiratory viruses—particularly on RSV—reinforces the necessity for integrated viral surveillance strategies.

## Figures and Tables

**Figure 1 viruses-17-01421-f001:**
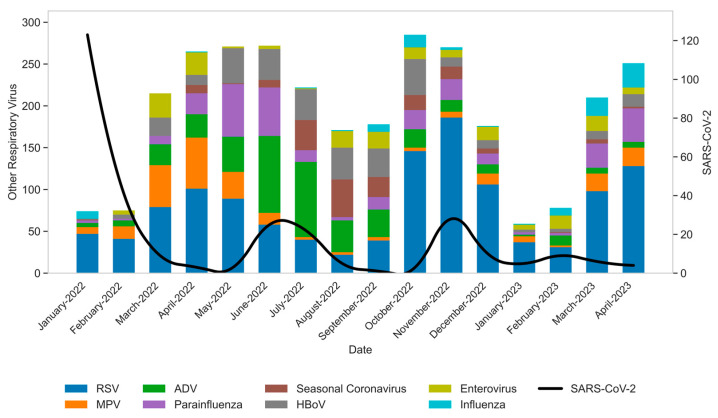
Distribution of total respiratory virus detected by real-time PCR from January 2022 to April 2023. SARS-CoV-2-positive samples are shown as a black line, while other respiratory viruses are presented as a colored bar.

**Figure 2 viruses-17-01421-f002:**
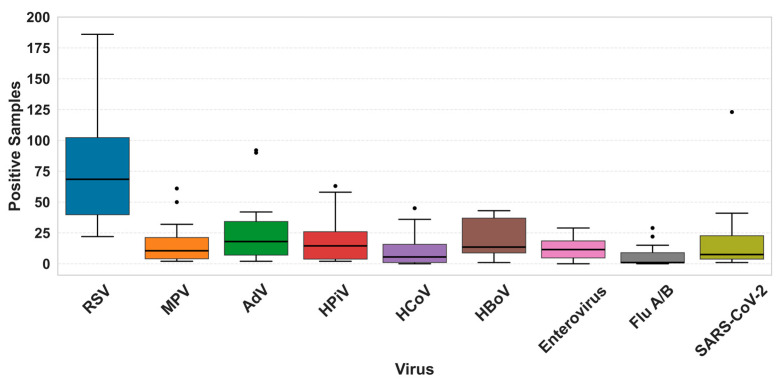
Distribution of positive samples for a single respiratory virus. Boxplots show the median, interquartile range, and dispersion in the number of positive detections for each virus. Outliers are indicated as individual points.

**Figure 3 viruses-17-01421-f003:**
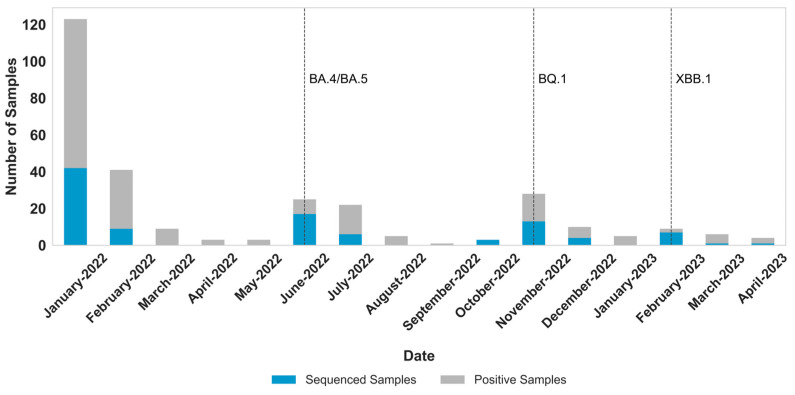
Distribution of positive SARS-CoV-2 samples per month (gray) with the distribution of sequenced samples (blue). The month of emergence of new lineages detected in this study is highlighted with dashed lines.

**Figure 4 viruses-17-01421-f004:**
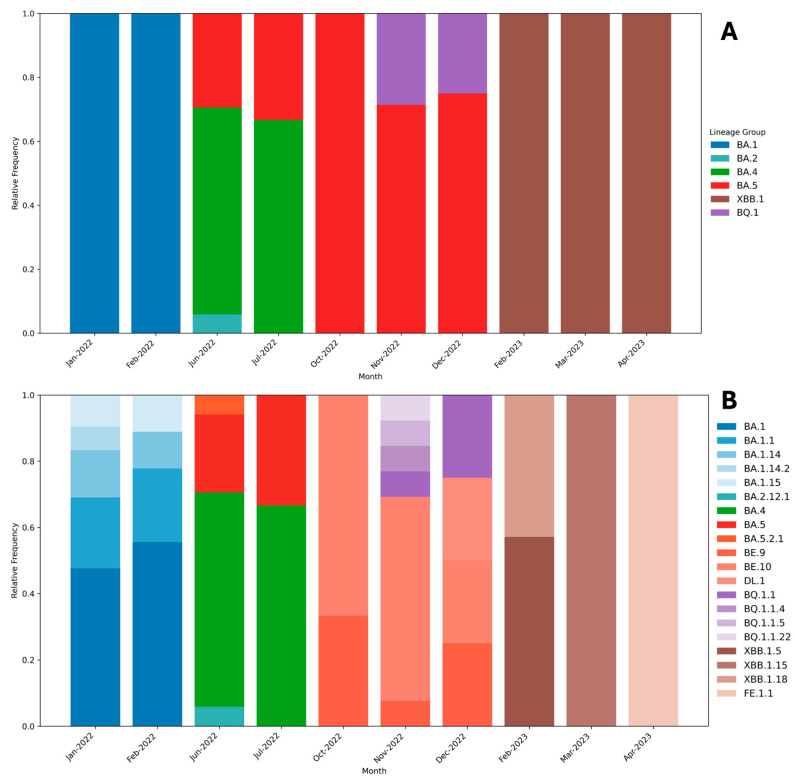
(**A**) Relative frequency and distribution of SARS-CoV-2 sublineages in pediatric patients detected in the study period. (**B**) Detailed distribution including all sublineages detected.

## Data Availability

Obtained SARS-CoV-2 sequences are available in GISAID database under the accession numbers: EPI_ISL_17959753 to EPI_ISL_17959844.

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
