# Peer review of "Genomic Diversity of SARS-CoV-2 Omicron Sublineages and Co-Circulation with Respiratory Viruses in Pediatric Patients in Sao Paulo, Brazil"

_viruses, 2025, doi:10.3390/v17111421_

Round 1

Reviewer 1 Report

Comments and Suggestions for Authors

Dorlass et al studied the distribution of Omicron variant of SARS-Cov-2 and its co-occurrence with other respiratory virus in Brazilian population. They studied the seasonal distribution of respiratory viruses and showed that RSV is the major infecting virus in children.

However, there are some concerns about the study that need to be addressed.

  1. Authors just sequenced and counted the numbers time-specifically. No information about new mutations, genomic diversity from sequencing data in this population etc, are extracted and presented.
  2. More insights about co-infections are lacking. For example, how many RSV positives are also SARS-CoV-2 positive and vice versa? Similarly, how many SARS-CoV-2 positives with each virus infection? These information with suitable interpretation are expected.
  3. In materials and methods, genomic analysis should be in more detail with mentioning all software.
  4. Statistical analysis would be written in more detail.
  5. Figure1 is not referred in the text until discussion (line 204). Should be mentioned in order with the text. The color of SARS-CoV-2 is almost not visible in Figure 1. Color code of SARS-CoV-2 could be changed.
  6. Table 1 does not have any heading.
  7. The labeling text in Figure 2 is illegible, and the font and resolution should be increased.

Author Response

Comments 1: Authors just sequenced and counted the numbers time-specifically. No information about new mutations, genomic diversity from sequencing data in this population etc, are extracted and presented
Response 1: Our manuscript aimed to investigate the temporal distribution and genomic diversity of SARS-CoV-2 Omicron sublineages in pediatric patients and to assess their impact on the circulation patterns of other respiratory viruses. By combining large-scale molecular surveillance and whole-genome sequencing of over 7,800 samples collected between January 2022 and April 2023, we identified the predominance of BA.1 and BA.5 sublineages and their influence on the epidemiological landscape of pediatric respiratory infections. Notably, we documented an atypical off-season RSV peak associated with changes in SARS-CoV-2 circulation and public health measures, revealing complex viral interactions during the Omicron period. These findings provide novel genomic and epidemiological insights with important implications for global viral surveillance, highlighting how SARS-CoV-2 evolution can reshape the seasonality of other respiratory viruses and reinforce the need for ongoing genomic monitoring and pediatric vaccination.

Comments 2: More insights about co-infections are lacking. For example, how many RSV positives are also SARS-CoV-2 positive and vice versa? Similarly, how many SARS-CoV-2 positives with each virus infection? These information with suitable interpretation are expected.
Response 2: Thank you for the suggestion. We agree with the proposed analysis, and have inserted it on our manuscript. Detailed results can be found in page 4, lines 152 - 164 (“…Co-infections involving RSV, AdV and HBoV were the most frequent, with RSV-HBoV presenting the most cases of co-infection (n = 66), followed by RSV/AdV (n = 61) and HBoV/AdV (n = 59). These co-infections accounted for 36.77% (n = 196) of the total number of detected co-infections. RSV/SARS-CoV-2 were the fourth most com-mon co-infection observed (n = 39, 7.31%). HCoV/Enterovirus, Flu/HPiV, Flu/RSV and MPV/Enterovirus were the least frequent co-infections detected during the study. Co-infections involving two viruses accounted for 472 (88.55%) of the total co-infections. The most common co-infection caused by three viruses detected is AdV/HBoV/RSV (n = 12, 2.25%). Co-infection with four viruses were rare, accounting for 5 cases in total (0.75%, Table S3). Cases peaked in June 2022 (n = 94) followed by October 2022 (n = 74). SARS-CoV-2 co-infections were most frequent in January 2022. However, June 2022 showed a higher number of types with SARS-CoV-2, such as RSV/SARS-CoV-2, HBoV/SARS-CoV-2, MPV/SARS-CoV-2, HPiV/SARS-CoV-2, AdV/SARS-CoV-2, HCoV/SARS-CoV-2 and HBoV/HPiV/SARS-CoV-2.…”) and discution in lines 238-249 (“…Co-infections overall exhibited a similar trend to single infections, with an increase in the number of samples with two or more viruses observed in the first and second semesters of 2022 (March and October). A peak in the number of co-infection cases was observed in June 2022 (n = 92), which coincided with a surge in AdV detections (Figure 1, Table S2). HBoV, AdV and RSV accounted for most co-infection cases (Table S3), which collaborates with other studies [31–33].…”). Supplementary material for co-infections was also provided as Supplementary Table 3 (Table S3). Updated text in the manuscript if necessary.

Comments 3: In materials and methods, genomic analysis should be in more detail with mentioning all software.

Response 3: Thank you for pointing this out. A more detailed methodology containing software parameters with citations can be found in the described methods, item 2.4 of page 3, lines 101-109 (“…Read files were generated with Torrent Suite v5.12 plugins on Ion Torrent S5 platform. Low-quality reads were removed from the analysis. Remaining reads were mapped directly to the SARS-CoV-2 reference genome (accession: MN908947) with TMAP. Coverage metrics were collected with CoverageAnalysis plugin (v5.10) set for a minimum of 50 reads depth as previously described [17]. Variants were called with variantCaller plugin (v5.12) with default “Germ Line-Low Stringency” parameters as seen here [18,19]. Called variants were annotated with COVID19AnnotateSnpEff (v1.0) and final consensus sequence was generated with GenerateConsensus plugin. SARS-CoV-2 lineages were identified with Pangolin software (v4.3.1) [20]…”). Updated text in the manuscript if necessary.

Comments 4: Statistical analysis would be written in more detail.

Response 4: We appreciate and agree with the suggestion, therefore a detailed statistical analysis used was inserted in the described methodology, page 3, item 2.5, lines 110-116 (“…Descriptive statistics were used for frequency and distribution analysis of detected respiratory viruses across sampling months. Associations between categorical variables, such as virus occurrence and sampling period, were evaluated using the Chi-square test of independence, with p-values < 0.05 were considered statistically significant. Data visualization and exploratory analysis were conducted with Python 3 using Pandas library (v2.1).…”). Updated text in the manuscript if necessary.

Comments 5: Figure1 is not referred in the text until discussion (line 204). Should be mentioned in order with the text. The color of SARS-CoV-2 is almost not visible in Figure 1. Color code of SARS-CoV-2 could be changed.

Response 5: Thank you for pointing out this error. We modified the text and Figure 1 is now correctly referred to in page 4 line 142. SARS-CoV-2 color in Figure 1 was changed as suggested. We also increase the thickness of the line in the figure for better visualization. Updated text in the manuscript if necessary.

Comments 6: Table 1 does not have any heading.

Response 6: Yes. Thank you for pointing this out. The table was submitted as supplementary table 2 (Table S2) as suggested by another reviewer and is no longer in the manuscript. Updated text in the manuscript if necessary.

Comments 7:  The labeling text in Figure 2 is illegible, and the font and resolution should be increased.
Response 7: Agree. The text was in fact illegible. We have altered the resolution of Figure 2 as well as the font size of the labels for better visualization. Updated text in the manuscript if necessary.

Reviewer 2 Report

Comments and Suggestions for Authors

In this study, the authors investigate the diversity and circulation patterns of SARS-CoV-2 Omicron sublineages in pediatric patients in Brazil, as well as assess their co-circulation with other respiratory viruses. The study is small but logically complete. The manuscript is well written, and the experimental section is clear. However, the text of the manuscript contains confusion regarding the numbering of figures and tables and the logic of the presentation of the material. After these deficiencies are corrected, the manuscript can be published in Viruses MDPI.

  1. In Ref. [1] there are no data on 700 million people and 7 million deaths worldwide caused by SARS-CoV-2, with over 700,000 in Brazil. Please add references to these digital data
  2. Lines 63-64: ….. BA.1, 63 BA.2, BA.5, BQ.1.1, XBB.1.16, EG.5.1, JN.1 and KP.2 [8,9]. Despite …..

a dot was missed

  1. Authors sometimes separate thousands with commas, sometimes not (for example, lines 108-109: 7335 and 3,369)
  2. On page 4 there is a table that, firstly, is not numbered (possible Table 2, Lines 160-161), and secondly, does not contain any new information.It is necessary to place in supplementary file. There is no Table 1 or means Table 2.
  3. there is no mention in the manuscript and no reference to Figure 2
  4. line 143: do the authors mean figure 3?
  5. Line 151-152: «RSV exhibit a higher dispersion in detected samples compared to other viruses in the study, ranging from a minimal of 31 cases to 186.»

Looking at Figure 2, one can say that the minimum value is 24, not 31.

  1. Lines 152-154: Overall, other respiratory viruses showed a low median value and narrower distribution in comparison to RSV, such as Flu A/B (Figure 2).
  2. lines 194-198: «Our study observed the expected increase in the number of RSV cases beginning in March 2022; but also notably documented an atypical second increase starting in September of the same year, reaching a pronounced peak in November, accounting for the highest number of RSV positive samples in 2022 (Table 1, Figure 1).» do the authors mean table 2? By the way, Table 1 is completely missing from the text
  3. Line 218: please use RSV for Respitarory Syncytial Virus.
  4. line 222: do the authors mean figure 2?

Author Response

Comments 1: In Ref. [1] there are no data on 700 million people and 7 million deaths worldwide caused by SARS-CoV-2, with over 700,000 in Brazil. Please add references to these digital data.
Response 1: Thank you for the suggestion. We agree that the cited reference did not contain the correct data. The reference for the data was inserted in page 1 line 53 (“…with over 700,000 in Brazil [1–3]…”). Updated text in the manuscript if necessary.

Comments 2: Lines 63-64: ….. BA.1, 63 BA.2, BA.5, BQ.1.1, XBB.1.16, EG.5.1, JN.1 and KP.2 [8,9]. Despite…..a dot was missed

Response 2: Thank you for pointing that out. The text was corrected in the revised manuscript in page 1 line 64 (“…such as BA.1, BA.2, BA.5, BQ.1.1, XBB.1.16, EG.5.1, JN.1 and KP.2 [10,11]. Despite the hig…”). Updated text in the manuscript if necessary.

Comments 3: Authors sometimes separate thousands with commas, sometimes not (for example, lines 108-109: 7335 and 3,369)

Response 3: Thank you for pointing this out. We agreed that inconsistencies were present in the manuscript. We corrected the use of commas in the reviewed manuscript, as can been seen in page 3 line 118. Updated text in the manuscript if necessary.

Comments 4:  On page 4 there is a table that, firstly, is not numbered (possibly Table 2, Lines 160-161), and secondly, does not contain any new information. It is necessary to place in the supplementary file. There is no Table 1 or means Table 2

Responde 4: Thank you for the suggestion. We agree that the numbering of tables was wrong and accept the suggestion of providing this data as a supplementary table (Table S2). Updated text in the manuscript if necessary.

Comments 5: there is no mention in the manuscript and no reference to Figure 2
Response 5: We agree. This mistake was corrected in the new manuscript. Figure 2 is correctly referenced in page 4 line 169. Updated text in the manuscript if necessary.

Comments 6: line 143: do the authors mean figure 3?

Response 6: Yes, thank you for bringing this error to light. We corrected the reference to the figure in page 4 line 171. Updated text in the manuscript if necessary.

Comments 7: Line 151-152: «RSV exhibits a higher dispersion in detected samples compared to other viruses in the study, ranging from a minimum of 31 cases to 186.» Looking at Figure 2, one can say that the minimum value is 24, not 31.

Response 7: Thank you for pointing this out. The referenced number was in fact wrong. The correct value was changed in the text in page 4 line 167 (“…ranging from a minimal of 22 cases to 186. Overall, other …”). Updated text in the manuscript if necessary.

Comments 8: Lines 152-154: Overall, other respiratory viruses showed a low median value and narrower distribution in comparison to RSV, such as Flu A/B (Figure 2).

Response 8:  Thank you for the suggestion. The corrected figure was referenced in the text in page 4 line 167 – 169 (“…Overall, other respiratory viruses showed a low median value and narrower distribution in comparison to RSV, such as FluA/B (Figure 2).…”).  Updated text in the manuscript if necessary.

Comments 9: lines 194-198: «Our study observed the expected increase in the number of RSV cases beginning in March 2022; but also notably documented an atypical second increase starting in September of the same year, reaching a pronounced peak in November, accounting for the highest number of RSV positive samples in 2022 (Table 1, Figure 1).» do the authors mean table 2? By the way, Table 1 is completely missing from the text

Response 9: Yes. Thank you for pointing out this error. We meant Table 2, and that table was submitted as a supplementary file (Table S2) as suggested. The correct table was referenced in the text as a supplementary table in Page 8 lines 209-213 (“…peak in November, accounting for the highest number of RSV positive samples in 2022 (Figure 1, Table S2).…”). Updated text in the manuscript if necessary.

Comments 10: Line 218: please use RSV for Respitarory Syncytial Virus.

Response 10: Thank you for the suggestion. The text was corrected in page 7 line 233 (“…abrupt compared to Respitarory Syncytial Virus. RSV demonstrated a sharp rise…”). Updated text in the manuscript if necessary.

Comments 11: line 222: do the authors mean figure 2?

Response 11: Yes. Thank you for pointing this out. The correct Figure was inserted in the text in page 7 line 237 (“…higher number of cases, underscores the distinct epidemiological dynamics of RSV throughout 2022 (Figure 2).…”). Updated text in the manuscript if necessary.

Round 2

Reviewer 1 Report

Comments and Suggestions for Authors

The paper is improved but the writings within Figure 2 and 3 is still illegible and should be improved. 

Author Response

Comments 1: The paper is improved but the writings within Figure 2 and 3 is still illegible and should be improved. 

Response 1: Thank you for all the valuable suggestions to improve our work. We completely agree that the text within the figures could be enhanced. We increased the font size in Figure 2 and changed the labels of both axes to bold.
 n Figure 3, we moved the legend to the bottom to allow for a larger font size within the figure. The fonts of both the x- and y-axes were also increased and set to bold for better visualization. All figures are provided at 600 dpi resolution. Updated Figures in the manuscript if necessary.

Reviewer 2 Report

Comments and Suggestions for Authors

the authors took into account the reviewer's suggestions

Author Response

Comments 1: the authors took into account the reviewer's suggestions

Response 1: Thank you for the suggestions. We believed that our paper have been improved by the reviewer suggestions.